# A Molecular Dynamics Study on the Dislocation-Precipitate Interaction in a Nickel Based Superalloy during the Tensile Deformation

**DOI:** 10.3390/ma16186140

**Published:** 2023-09-09

**Authors:** Chang-Feng Wan, Li-Gang Sun, Hai-Long Qin, Zhong-Nan Bi, Dong-Feng Li

**Affiliations:** 1School of Science, Harbin Institute of Technology, Shenzhen 518055, China; 18B358027@stu.hit.edu.cn (C.-F.W.); sunligang@hit.edu.cn (L.-G.S.); 2Beijing Key Laboratory of Advanced High Temperature Materials, Central Iron and Steel Research Institute, Beijing 100081, China; hailongqin@126.com (H.-L.Q.); bizhongnan21@aliyun.com (Z.-N.B.)

**Keywords:** precipitate strengthening, molecular dynamics, dislocation-precipitate interaction, anisotropy

## Abstract

In the present paper, the dislocation-precipitate interaction in the Inconel 718 superalloy is studied by means of molecular dynamics simulation. The atomistic model composed of the ellipsoidal $Ni_{3}Nb$ precipitate (γ″ phase) and the Ni matrix is constructed, and tensile tests on the composite $Ni_{3}Nb@Ni$ system along different loading directions are simulated. The dislocation propagation behaviors in the precipitate interior and at the surface of the precipitate are characterized. The results indicate that the dislocation shearing and bypassing simultaneously occur during plastic deformation. The contact position of the dislocation on the surface of the precipitate could affect the penetration depth of the dislocation. The maximum obstacle size, allowing for the dislocation shearing on the slip planes, is found to be close to 20 nm. The investigation of anisotropic plastic deformation behavior shows that the composite system under the loading direction along the major axis of the precipitate experiences stronger shear strain localizations than that with the loading direction along the minor axis of the precipitate. The precipitate size effect is quantified, indicating that the larger the precipitate, the lower the elastic limit of the flow stress of the composite system. The dislocation accumulations in the precipitate are also examined with the dislocation densities given on specific slip systems. These findings provide atomistic insights into the mechanical behavior of nickel-based superalloys with nano-precipitates.

## 1. Introduction

Inconel 718 (IN718), a precipitate-strengthened nickel-based superalloy, has been widely used in aerospace, petrochemical, and nuclear industries due to its exceptional mechanical properties [1,2,3,4]. With a yield strength of approximately 1050 MPa, IN718 exhibits high strength and excellent performance across a wide temperature range, from cryogenic temperatures up to 920 K [5]. Its remarkable combination of strength, toughness, and fatigue resistance makes it suitable for demanding applications. Furthermore, IN718 demonstrates excellent resistance to oxidation and corrosion, ensuring its longevity and reliability in harsh environments. The alloy’s ability to maintain its mechanical properties under extreme conditions has solidified its reputation as a highly reliable and versatile material. The alloy has a quite complex nanostructure, which involves two kinds of nano-precipitates, namely $\gamma^{\prime }$ and $\gamma^{\prime \prime }$ phases, to affect the dislocation movement in the γ matrix phase [6,7,8]. The $\gamma^{\prime \prime }$ precipitate (ellipsoidal and disc-like) is the main strengthening precipitate and has a coherent $DO_{22}$ structure with a volume fraction of approx. 13–15%. The spherical $\gamma^{\prime }$ precipitate has a volume fraction of approx. 3–5% and is less relevant than the $\gamma^{\prime \prime }$ precipitate from the strengthening point of view. There is also a small amount of needle-like δ phase [9], which can be transformed from the $\gamma^{\prime \prime }$ phase and is usually located at grain boundaries to stabilize the grain morphology. Knowledge of the underlying mechanisms of how dislocations interact with the complex precipitate system of IN718 is crucial to understanding the role that the nano-precipitates play on the mechanical properties of IN718.

There are two dislocation-precipitate interaction mechanisms in IN718, namely dislocation shearing and dislocation bypassing [7,8,10,11]. It has been known that the competition between the two mechanisms depends largely on the precipitate size [10,11,12,13] such that the mobile dislocations likely shear through the smaller precipitates and bypass the larger ones. The critical radius of the precipitates is often used to indicate the transition from the dislocation shearing to the dislocation bypassing [8,11]. However, the ellipsoidal geometry of $\gamma^{\prime \prime }$ precipitate has not been well accounted for. Molecular dynamics (MD) simulations are widely used to explore the lattice-scale physics of the dislocation-precipitate interaction [14,15,16,17,18,19]. Bacon, D.J. and Osetsky, Y.N. [14] studied the interaction between the edge dislocation and copper precipitates in α-iron alloy, quantitatively revealing the dependence of critical resolved shear stress (CRSS) on temperature, precipitate size, and the mismatch level. It is found that the smaller the size of the copper precipitates, the easier the dislocation shearing. Similar results have been obtained for other alloys, e.g., medium- and high-entropy alloys [15,16,18], Mg-Al alloy [17] and Mg-Zn alloy [19]. The MD method can also be used to study the dependence of the plastic flow and the dislocation motion on the size of nano-pore or nano-particle in copper [20]. Apart from the atomistic insights into the dislocation-precipitate interaction achieved, the MD modeling can also play a fundamental role in the multi-scale modeling, for example a combination of the MD method and discrete dislocation dynamics (DDD) model, to explore the evolution of dislocation patterns with the effect of precipitate morphology taken into account explicitly [21,22,23,24,25,26]. In these modeling studies, MD simulation is typically used to gain some material parameters at the atomic level and to support the higher-level DDD simulations. Although the modeling methods at atomic levels, such as MD, DDD, and their combination, have been illustrated as a powerful tool to fundamentally quantify the interplay between dislocations and precipitates in a wide range of alloys, explicit atomistic studies on the dislocation shearing and bypassing in IN718 are still limited.

In this work, MD studies are performed to investigate the interaction between the dislocations and the $\gamma^{\prime \prime }$ precipitates in the IN718 superalloy. The stress-driven motion of dislocations and its effect on the mechanical properties are focused on. The morphological features of the $\gamma^{\prime \prime }$ are explicitly considered in the MD model, and the uniaxial tensile tests are simulated along different loading directions. A series of snapshots on the microstructure evolution with the dislocation shearing and bypassing going through in the precipitate are given. Moreover, the anisotropic effect and the precipitate size effect are investigated. The paper is structured as follows: In Section 2, the details of the MD modeling are introduced. Section 3 shows the results of the MD simulation, followed by the discussion of the results in Section 4, and Section 5 presents the concluding remarks.

## 2. Molecular Dynamics Model

Figure 1 illustrates the molecular dynamics (MD) model employed in this study. A cubic unit cell, measuring 25 × 25 × 25 nm${}^{3}$, is constructed consisting of a face-centered cubic (FCC) Ni matrix. At the center of the cell, a disk-like single-crystalline $Ni_{3}Nb$ particle with a $DO_{22}$ structure is coherently embedded within the Ni matrix. The major axis of the ellipsoidal $Ni_{3}Nb$ precipitate is set to 20 nm, while the minor axis is 10 nm. To investigate the interaction between dislocations and the $Ni_{3}Nb$ precipitate, a pair of dislocations is pre-set within the pure Ni matrix. The total number of atoms in the unit cell is approximately 1,300,000. Due to the complex composition of IN718, and the lack of a reliable force field for simulating its atomic interactions, pure Ni is assumed to represent the matrix material in this study. Pure Ni and IN718 have similar stacking fault energies, which govern the evolution of crystalline defects such as dislocations and stacking faults during plastic deformation. The atomic interactions are described using the embedded-atom-method (EAM) potential developed by Zhang, Y. et al. [27]. Periodic boundary conditions are applied to the MD model, and the simulations are performed using the LAMMPS [28] package. The model is first relaxed at 300 K for 500 ps under a Nose-Hoover thermostat [29] and 0 bar external pressure. Subsequently, uniaxial tensile tests are simulated at a constant strain rate of $2.5\times 10^{8}\;$ s${}^{-1}$ and a temperature of 300 K. Tensile tests are conducted with loading along the [010] direction (major axis of the precipitate) and the [001] direction (minor axis of the precipitate), which represent loading parallel and perpendicular to the disk plane, respectively. Local atomic shear strain is calculated to investigate plastic shearing during the tensile deformation [30]. To characterize the crystal structure, the common neighbor analysis (CNA) method is employed [31,32,33], with green atoms representing the FCC structure, red atoms representing the hexagonal close-packed (HCP) structure, and white representing grain boundaries, dislocations, and other disordered atoms. The dislocation extraction algorithm (DXA) [33,34], integrated into the OVITO 3.7.8 software [35], is utilized to determine the dislocation lines within the simulation.

## 3. Results

### 3.1. Dislocation-Precipitate Interaction Simulations

When a single dislocation is initially introduced into the unit cell, it will undergo movement and evolution under the influence of the driving force during tensile deformation, leading to the emergence of additional new dislocations. Figure 2 shows the distribution of dislocations in the unit cell at a tensile strain of 6.25%. Here, the matrix is set transparent, and the dislocation lines and the atoms of the precipitate are visible. Figure 2a,b shows the views from different directions, with the precipitate viewed as an ellipsoid and a disk, respectively. It can be seen that a large number of dislocations surround the precipitate (marked by the black arrows). Quite a few dislocations can be seen on the surface of the precipitate (marked by the red arrows), which are penetrating into the precipitate. This could be the direct evidence of the dislocation shearing at this applied strain level. In order to examine whether the dislocation bypassing occurs or is ongoing, the unit cell and its periodic counterpart attached on the {010} plane are visualized in Figure 2c. Some bowed dislocation lines (marked by black arrows) can be seen between the two precipitates, probably indicating the dislocation bypassing based on the Orowan looping concept [36]. It is important to note that when the two unit cells are attached on the {001} plane, no dislocation lines can be detected. However, if the periodic unit cells are tiled throughout the entire space, it is possible to observe more curved dislocation lines. It should be noted that mobile dislocations tend to move along slip planes in the direction of shear stress under the influence of applied stress. Ideally, the Orowan bowing would be observed on the planes where the dislocation glides. However, since MD simulations are used here to capture the dynamic 3D configurations of dislocations, it is hard to visualize the dislocation lines on a specific plane.

In order to further examine the dislocation shearing phenomenon, the atomic configurations of the precipitate are focused on. Figure 3 shows the atoms of the precipitate at different applied strains. It can be observed that when a dislocation shears into the precipitate, a stacking fault is generated in the region where the dislocation sweeps through. This creates an easier path for subsequent dislocations to shear into the precipitate along the same path [6]. In addition, given in the figure are the identified slip systems (defined in Figure 3) for the dislocation shearing. Figure 3c,d shows that the dislocations shearing into the precipitate can be on the same slip plane but along different slip directions.

### 3.2. Atomic Shear Strain Distributions during the Tensile Deformation

In order to further probe the local deformation at the atomic level, a series of atomic configurations showing the local equivalent atomic shear strain in the precipitate under different applied strains are presented in Figure 4. In Figure 4a–c, most atoms experience almost no shear strain (blue), and only a few atoms experience local shear strain (with white and red colors). It can be seen in Figure 4a–c that atomic shear strain tends to occur locally on some lines on the surface of the precipitate. These lines represent the path of slip planes on the surface of the precipitate. As the applied strain increases, more trace lines become visible due to the activation of additional slip planes for the dislocations to glide on. Certain slip planes are preferred for dislocations, which can be identified by the relatively high atomic shear strains observed on their corresponding trace lines. To provide a clear view of the shear strain distribution within the precipitate, Figure 4d–f shows the shear strain distributions over the atoms in the precipitate, with most of the blue atoms neglected. Four regions corresponding to four slip planes are marked in the figure. It can be seen that the atoms on the slip plane ‘C’ indicated by black arrows experience relatively low shear strain. This implies that the precipitate is hard to shear at the trace line position on this slip plane. For the atoms on the slip plane ‘B’, relatively high shear strain is experienced, implying that the dislocation shearing is prone to taking place. Note that the trace line of the slip plane ‘C’ is closer to the center of the precipitate than that of the slip plane ‘B’. The atoms with shear strain experienced on the other two slip planes can also be seen. These results indicate that the dislocation shearing may occur all over the precipitate.

In order to compare the atomic shear strains in the $Ni_{3}Nb$ precipitate with that in the pure Ni matrix, Figure 5 shows the shear strain on the two mid-planes vertical to the [001] and [100] directions, respectively, at 12.5% tensile strain. The boundary of the precipitate is outlined by the white lines. Uneven plastic shear can be observed in the figure, with the matrix undergoing more severe deformation compared to the precipitate. Furthermore, the dislocation is unable to easily cut through the central area of the precipitate, leading to a reduced ability of the precipitate to accommodate plastic deformation.

### 3.3. Anisotropic Mechanical Responses

To examine the anisotropic behavior of the unit cell, the atomic shear strain distributions of the mid-plane at a tensile strain of 25% along two loading directions are investigated and shown in Figure 6, where the precipitates are indicated by the dashed lines. For both cases, strong localizations of the atomic shear strain can be seen in the matrix phase, with slight strain localizations in the precipitate. It can be observed from the figure that the precipitate with the loading direction parallel to the disk plane undergoes more plastic deformation compared to the one with the loading direction perpendicular to the disk plane. This suggests that dislocation shearing is more likely to occur when the loading direction aligns with the disk plane of the γ″ precipitate. This trend is not surprising, as the shape of the precipitate (ellipsoidal and disc-like) has a significant influence on dislocation shearing behavior. Additionally, Figure 6 presents the lengths of the precipitate axes and the corresponding half spacing between the two periodic precipitates in both the deformed configuration and the initial configuration. It demonstrates strong anisotropic deformation in both the matrix and the precipitate. Specifically, when the loading direction aligns with the major axis of the precipitate, the axis is stretched by 8.5%, while when the loading direction aligns with the minor axis of the precipitate, the axis is stretched by 3%.

### 3.4. The Effect of Precipitate Size on the Stress–Strain Response

In order to examine the precipitate size effect, the ratio between the major and the minor axes of the precipitate is fixed to 2. Additional four-unit cells are constructed with the half length of the major axis varying from 15 nm to 40 nm, and tensile tests along different loading directions are simulated. Figure 7a,b shows the uniaxial tensile stress–strain curves with different precipitate sizes and under different loading directions. When the unit cells undergo elastic deformation, the overall stresses along the tensile direction increase linearly with the applied strain. However, once the applied strain exceeds certain thresholds, strongly nonlinear stress–strain responses occur due to plastic deformation controlled by dislocation movements. After surpassing the elastic limits, the stress drops dramatically at first and then fluctuates. This nonlinear behavior is attributed to the dynamic nature of dislocations and the complex interaction between dislocations and precipitates. As expected, the size of the precipitate has a significant effect on the elastic limit of the flow stress, with larger precipitates leading to lower elastic limits of the flow stress. It can be seen in Figure 7a,b that the difference of the flow stresses at the elastic limit for 10 nm and 40 nm cases when the loading direction is parallel to the disk plane is approx. 7 GPa. When the loading direction is vertical to the disk plane, the difference is about 6 GPa. In addition, the amplitude of the stress fluctuations depends on the loading directions. In order to clearly see the precipitate size effect, Figure 7c shows the average flow stress vs. the half length of the major axis with error bars included. It is evident that as the size of the precipitate increases, the average flow stress decreases. Additionally, the unit cell with the loading direction perpendicular to the disk plane exhibits a higher average stress compared to the unit cell with the loading direction parallel to the disk plane. This difference becomes more prominent as the half length of the major axis increases from 10 nm to 30 nm. However, this trend does not hold for the case of a 40 nm precipitate.

## 4. Discussion

### 4.1. The Critical Precipitate Size for Dislocation Shearing

In the classical dislocation-precipitate interaction theory, the dislocation prefers to shear the precipitate if the precipitate size is small enough, otherwise, dislocation tends to bypass it with the so-called Orowan loop (see e.g., [37,38]) generated. The classic theory fits well to the cases with spherical precipitates [39,40]. To quantify the interaction mechanisms in the alloys containing non-spherical precipitates, the mean radius of the precipitate has been used [8,11]. The mean radius refers to the radius of a hypothetical spherical precipitate that has the same volume as the actual precipitate. However, the present paper shows that the interaction between the dislocation and the ellipsoidal γ″ precipitate in the IN718 superalloy is quite complex, and both dislocation shearing and bypassing may be induced simultaneously. The dislocation shearing and bypassing with regard to the γ″ precipitate can be illustrated as follows.

Figure 8 illustrates potential interactions between dislocations and precipitates. In Figure 8a, the dislocation line bypasses the precipitates aligned along the major axis. In Figure 8b, the dislocation line intends to shear into the precipitates aligned along the minor axis. The size of the obstacle on that plane can have a significant impact on the dislocation’s behavior. In the case of dislocation bypassing, as shown in Figure 8a, the obstacle size refers to the length of the major axis of the precipitate. In the other case shown in Figure 8b, the obstacle size is defined by the length of the minor axis of the precipitate. Therefore, the maximum size of the γ″ phase for dislocation shearing can be estimated to be in the interval from 10 nm (the length of the minor axis) to 20 nm (the length of the major axis). This estimation is close to the prediction proposed in [11], where the critical size for dislocation shearing is approx. 18 nm. It should be noted that there exist other estimations of the critical precipitate size for dislocation shearing in the literature [8,41], ranging from 40 nm to 120 nm.

To further verify the estimated critical precipitate size for dislocation shearing, the dislocation glide behavior in the precipitate with a major axis radius of 40 nm and a minor axis radius of 20 nm is investigated. Figure 9a,b shows the atomic shear strain distribution in the precipitate and the matrix under 12.5% tensile strain for the unit cell. It can be seen that, similar to the results in Figure 5, the high atomic shear strains are mainly located in the edge region of the precipitate. The estimated regions where dislocation shearing and bypassing could take place in the precipitate are given in Figure 9c,d. Here, the red regions are obtained based on the estimated critical obstacle size (20 nm here). It can be seen that the red regions in Figure 9c,d are qualitatively consistent with the high atomic shear strain areas in Figure 9a,b. Figure 9e,f shows the estimated dislocation shearing regions in the precipitate with a major axis radius of 20 nm and a minor axis radius of 10 nm. It can be seen that the portion of the red regions is bigger than that in Figure 9c,d, and it fits well with the simulated results shown in Figure 5.

### 4.2. Dislocation Accumulations in the Precipitate

From a mechanistic point of view, the dislocation density in the precipitate is a key state variable to reflect the precipitate strengthening [42,43]. However, accurately measuring or estimating dislocation density levels within the γ″ precipitate of IN718 superalloy can be challenging due to the small size of the precipitates. MD simulations offer a convenient and reliable way to investigate the evolution of dislocations within the precipitate. It can be observed in Figure 3 that when dislocations shear into the precipitate, certain slip systems are favored. To quantify the dislocation accumulations along different slip systems, the dislocation densities in the precipitate are statistically calculated for the unit cell with a standard precipitate as shown in Table 1. According to the symmetry of the slip directions, twelve slip systems are categorized into three groups.The slip systems in Group 1 are the slip systems with the slip direction perpendicular to the [100] direction (containing the slip systems, D1, C1, A2, and B2 as shown in Figure 3f). The slip systems D4, B4, A3, and C3 are in Group 2 with the slip direction perpendicular to [010]. The slip systems, A6, B5, C5, and D6 are in Group 3 with the slip direction perpendicular to [001]. Table 1 shows the dislocation densities at a tensile strain of 6.0% for the case with the loading direction along the major axis of the precipitate. It shows that the slip systems in Group 3 are most preferred by the dislocations to accumulate along the precipitate. The slip systems in Group 3 are particularly favored by dislocations in the precipitate. When dislocations attach to the precipitate along the slip systems in Group 3, they form relatively short trace lines. This arises because the slip directions are perpendicular to the minor axis, making the obstacle size on the slip plane relatively small. With a small obstacle size, the dislocations can easily cut through the precipitate along the slip systems where there is a higher density of dislocations. This phenomenon elucidates the anisotropy of atomic shear strain. In theory, the active direction of slip cannot be perpendicular to the direction of loading. Consequently, when loading runs parallel to the plane of the disk, it can activate the slip systems in Group 3 and Group 1, whereas loading vertically to the disk plane can activate the slip systems in Group 1 and Group 2. Given the preference for specific slip systems demonstrated in Table 1, loading parallel to the disk plane could induce more dislocation motion within the precipitate. Greater dislocation motion corresponds to a heightened atomic shear strain.

## 5. Conclusions

In this work, the MD simulations are performed to investigate the dislocation-precipitate interaction in the IN718 superalloy. The main findings and conclusions are summarized as follows.

(1)The present paper explicitly examined the dislocation movements within and around the ellipsoidal γ″ nano-precipitate. Both dislocation bypassing and shearing are identified for the γ″ precipitate. This underscores the significant influence of the dislocation’s attachment point on the precipitate’s surface on the depth to which the dislocation can penetrate. Specifically, dislocations tend to shear into the precipitate from positions where the cross-section of the precipitate intersected by the active slip planes is relatively small. The maximum value of such a size, permitting dislocation shearing, is estimated to fall within the range of 10 nm to 20 nm, with a close approximation to 20 nm. This estimation of the precipitate’s critical size aligns well with the simulation outcomes.(2)The analysis of dislocation accumulation demonstrates that, in scenarios where dislocations shear into the precipitate with a loading direction parallel to the major axis of the precipitate, all twelve FCC slip systems contribute to accommodating a certain quantity of dislocations. Among these slip systems, the dislocation with a slip direction perpendicular to the disk plane is particularly favored by exhibiting the highest dislocation density. These dislocations encountered obstacles of relatively small size during their motion.(3)The atomic shear strain is uneven and anisotropic. It tends to localize around the interface between the matrix and the precipitate. The unit cell with the loading direction parallel to the major axis of the precipitate experiences stronger strain localizations than that with the loading along the minor axis of the precipitate. In addition, the flow stress fluctuations for the unit cell with the loading direction parallel to the major axis are stronger than that under the loading along the minor axis of the precipitate. This anisotropy arises from the anisotropic dislocation slip in the ellipsoidal precipitate. More precisely, the dislocations with a slip direction perpendicular to the disk plane experience lower resistance as they move within the precipitate.(4)The investigation into the precipitate size effect reveals that, by maintaining a constant volume fraction of precipitate, larger precipitates correspond to reduced elastic limits of flow stress. In the case of sizable γ″ precipitates, plastic deformation within the precipitate is constrained to the limited edge regions. This limitation stems from the fact that slip resistance within the γ″ phase escalates from the edges to the center of the precipitate. In essence, shearing across the center of the precipitate necessitates a relatively substantial driving force that can overcome the heightened slip resistance posed by the larger obstacle size.

The present paper focuses on the dislocation-precipitate interaction and its implications on the mechanical behavior of the superalloy. A detailed analysis of the corresponding atomistic mechanisms is performed. The findings are expected to be beneficial for the development of high-performance superalloys by rationally utilizing nano-precipitation.

## Figures and Tables

**Figure 1 materials-16-06140-f001:**
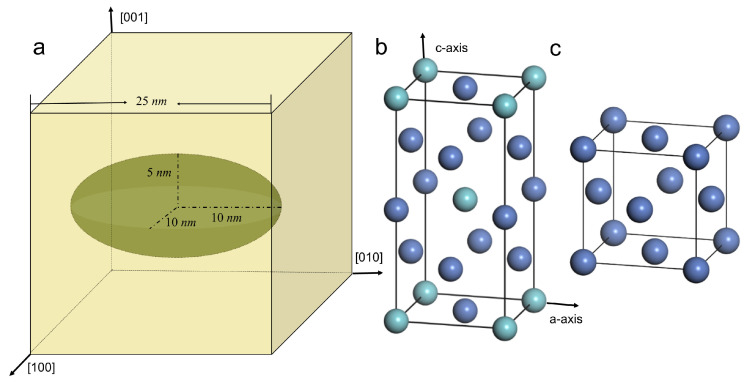
Illustration of the MD model with (**a**–**c**) showing the morphology of the unit cell, the atomic $DO_{22}$ structure of the precipitate, and the fcc structure of the matrix, respectively. In (**b**,**c**), the cyan ball represents Niobium atoms and the blue represents Nickel atoms.

**Figure 2 materials-16-06140-f002:**
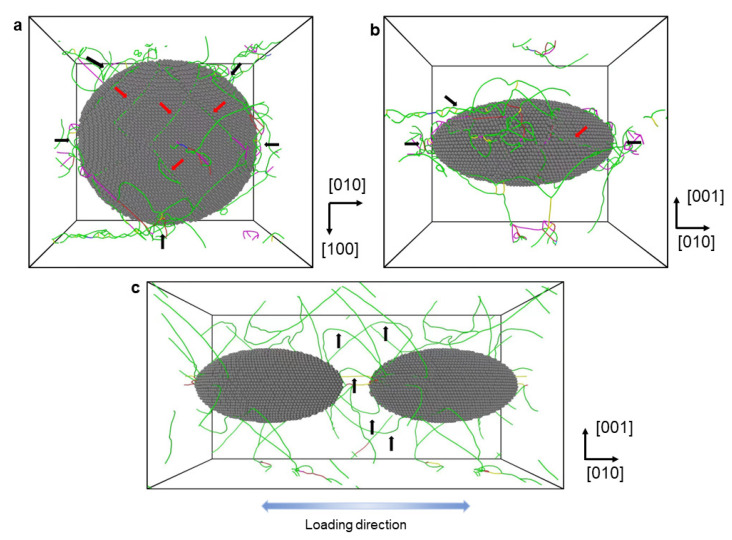
The distributions of dislocation lines in the unit cell at a tensile strain of 6.25%. The pure Ni matrix atoms are set transparent. The views of the unit cell, (**a**) from the [001] and (**b**) from the [100] directions are presented here. The view (**c**) from the [100] direction for the unit cell with its periodic duplication along the [010] direction is also presented. In these figures, the black arrows indicate the dislocations surrounding or looping the precipitates, while the red arrows mark the dislocations which are penetrating the precipitate.

**Figure 3 materials-16-06140-f003:**
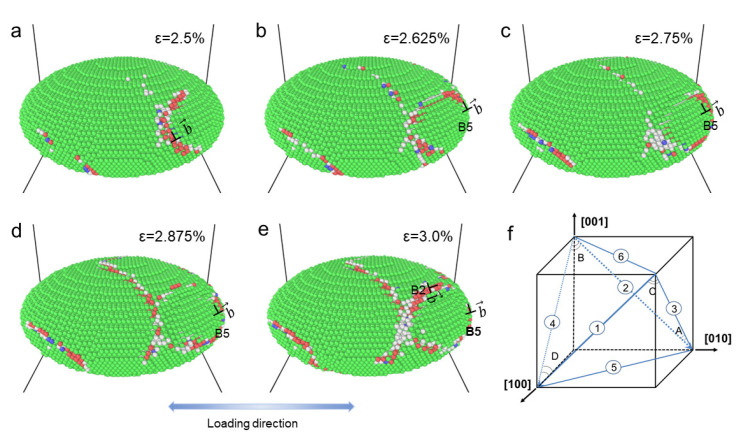
The snapshots of atomic configurations (CNA method) during the uniaxial tensile deformation of the $Ni_{3}Nb$ system sample, at tensile strains of (**a**) 2.5%; (**b**) 2.625%; (**c**) 2.75%; (**d**) 2.875%; (**e**) 3.0%. The ⊥ represents dislocations and $\vec{b}$ denotes burgers vector. The slip system illustration (**f**) for the FCC crystal is also presented. The green dots represent $Ni_{3}Nb$, while the red dots represent the HCP structure. The white dots represent boundaries and other defects.

**Figure 4 materials-16-06140-f004:**
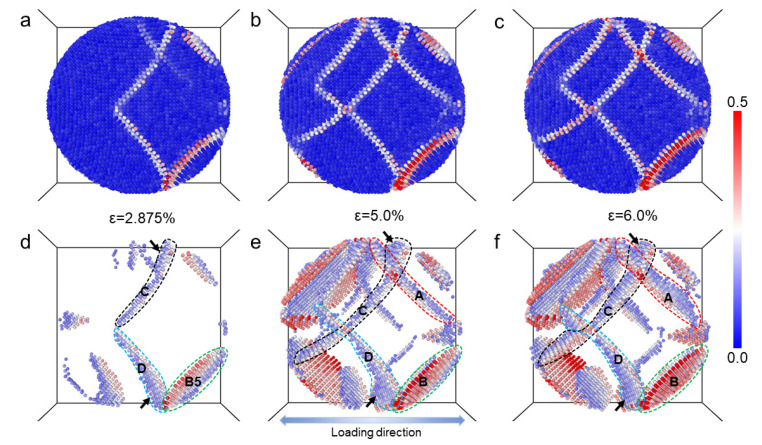
The snapshots of atomic configurations colored by atomic shear strain, at tensile strains of (**a**) ϵ = 2.875%, (**b**) ϵ = 5.0% and (**c**) ϵ = 6.0%. In the according atomic configurations (**d**–**f**), the atoms with the an atomic shear strain of less than 0.15 are set invisible. The regions marked by the dashed lines highlight the atoms on some slip planes.

**Figure 5 materials-16-06140-f005:**
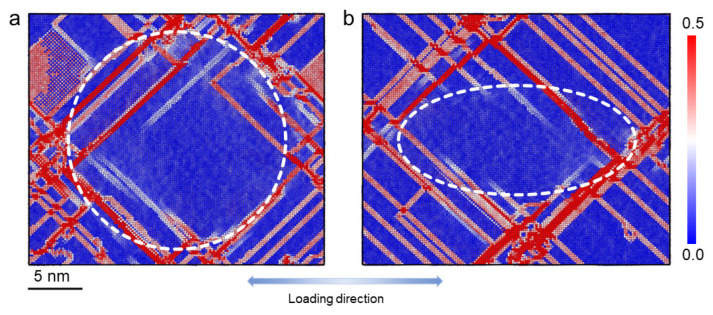
The atomic configurations of two mid-planes of the sample at a tensile strain of 12.5% (colored by atomic shear strain), (**a**) The mid-plane with the circular cross-section of the precipitate and (**b**) The mid-plane with the elliptical cross-section of the precipitate. The region of precipitate is highlighted by the white dashed lines.

**Figure 6 materials-16-06140-f006:**
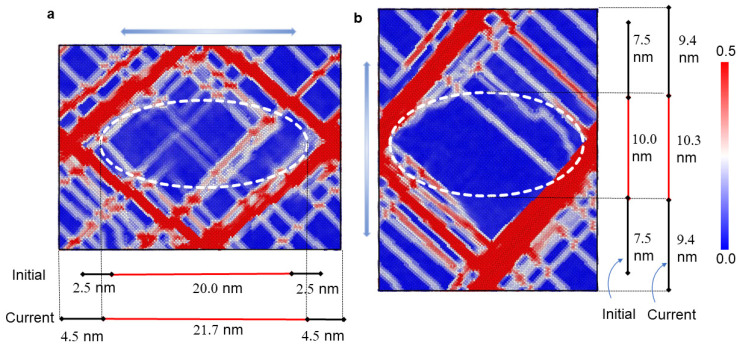
Atomic shear strain distributions in the mid-plane vertical to [100] at 25% tensile strain in (**a**) [010] and (**b**) [001] directions.

**Figure 7 materials-16-06140-f007:**
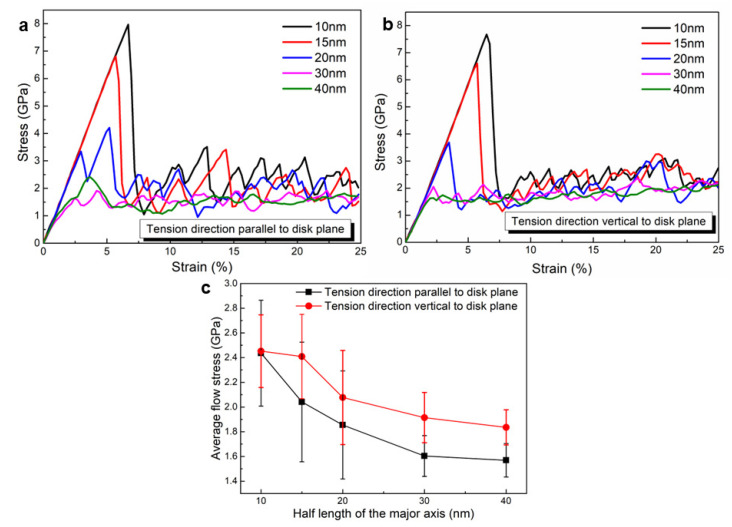
The size effect and loading direction dependent mechanical response in terms of (**a**,**b**) stress–strain curves with different precipitate sizes and loading directions and (**c**) the average flow stress obtained from (**a**,**b**).

**Figure 8 materials-16-06140-f008:**
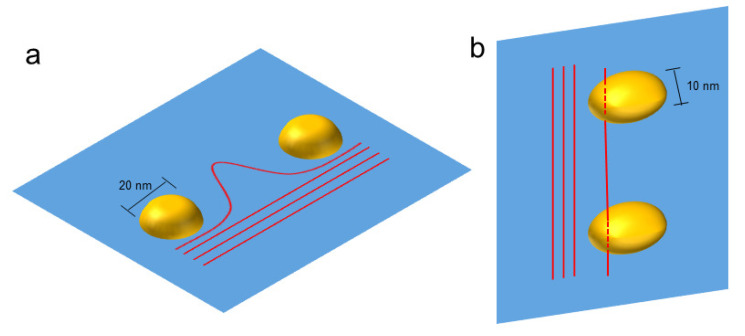
The illustrations of the dislocation-precipitate interaction with (**a**) the dislocation bypassing the two precipitates which are aligned along the major axis and (**b**) dislocation shearing into the two precipitates which are aligned along the minor axis.

**Figure 9 materials-16-06140-f009:**
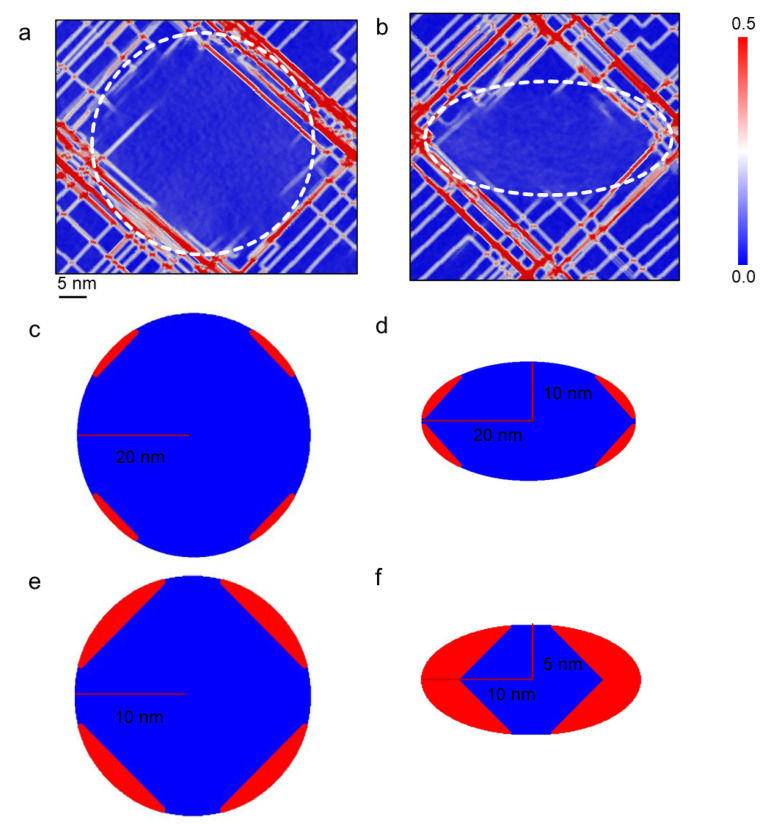
The atomic shear strain distributions of two mid-planes of the sample (the length of the major axis of the precipitate is 80 nm) at a tensile strain of 12.5% with (**a**) The mid-plane with the circular cross-section of the precipitate and (**b**) The mid-plane with the elliptical cross-section of the precipitate. The precipitate-occupied region is highlighted by the white dashed lines. The according illustrations of the dislocation shearing regions are also given based on the estimated critical obstacle size with (**c**,**d**) for large precipitate and (**e**,**f**) for standard precipitate.

**Table 1 materials-16-06140-t001:** Statistical dislocation density in the precipitate when ε=6.0%.

Group 1	Group 2	Group 3
**(C1, D1, A2, and B2)**	**(A3, C3, B4 and D4)**	**(B5, C5, A6 and D6)**
$2.87\times 10^{11}\;{\mathrm{mm}}^{-2}$	$3.30\times 10^{11}\;{\mathrm{mm}}^{-2}$	$5.84\times 10^{11}\;{\mathrm{mm}}^{-2}$

## Data Availability

The data presented in this study are available on reasonable request from the corresponding author.

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
