# Peer review of "A Molecular Dynamics Study on the Dislocation-Precipitate Interaction in a Nickel Based Superalloy during the Tensile Deformation"

_materials, 2023, doi:10.3390/ma16186140_

Round 1

Reviewer 1 Report

The paper seems as an exercise in computation using the LAMMPS software utilizing EAM potential to model Ni-based superalloy, and then the results are just visualized or analyzed using the software/analysis tools provided in Ovito. Although MD simulation has been conducted, the paper fails to provide any deep atomistic insight (which could benefit the research community) besides stating obvious facts in the conclusion such as

"when the loading direction is parallel to the precipitate major axis, the unit cell experiences stronger strain localisations and flow stress fluctuations are stronger compared to loading along the minor axis direction."

The research seems to be incremental and the paper reports observations that are not surprising. In my opinion, the manuscript is more suitable for a  specific (archival) journal, and I do not recommend publication in Materials.

The authors can work on word choices and sentence construction to describe their interpretations.

Author Response

Dear reviewer,

We greatly appreciate your valuable feedback on the manuscript. Our team has diligently revised the document, taking into account your insightful suggestions. Throughout the manuscript, we've incorporated specific revisions, and these are readily identifiable through highlighted text. In particular, red highlights signal reorganization that enhances both readability and logical coherence, while words added based on your comments are marked in a distinctive blue font. 

In light of concern about the quality of English writing, we've conducted a comprehensive review of the manuscript. We've focused our efforts on clarifying paragraphs that may not have been expressed with clarity. Such revisions are indicated by the presence of red font highlights in the manuscript.

Here are explanations addressing your comments.

  1. Although MD simulation has been conducted, the paper fails to provide any deep atomistic insight (which could benefit the research community) besides stating obvious facts in the conclusion such as

"When the loading direction is parallel to the precipitate major axis, the unit cell experiences stronger strain localisations and flow stress fluctuations are stronger compared to loading along the minor axis direction."

Response to comment: We appreciate the reviewer's valuable insights and have taken measures to enhance this facet of our work. Specifically, we've focused on refining the methodology, conclusion, and certain parts of the results section to underscore the importance of our findings.

We want to underscore the fact that the interplay between dislocations and precipitates yields these atomistic outcomes. Firstly, we've addressed the variance in slip resistance induced by the ellipsoidal precipitate in twelve distinct slip systems, which in turn causes discrepancies in dislocation density across these systems (as elucidated in the second conclusion of the conclusion section). Secondly, we've placed added emphasis on how the ellipsoidal precipitate significantly impacts dislocation slip resistance, ultimately leading to the emergence of anisotropic atomic shear strain (as elucidated in the third conclusion of the conclusion section). Lastly, our efforts have been directed toward unraveling the size effect through an exploration of the interaction between dislocations and precipitates. We emphasize that the distribution of atomic shear strain becomes particularly pronounced within the surface region of the precipitate when dislocations traverse it. This phenomenon occurs within a critical obstacle size range, which serves as the boundary between dislocation shearing and bypassing.

While we acknowledge the accuracy of your comments, our aim remains to contribute valuable insights to the broader research community regarding the ways in which an ellipsoidal precipitate can wield influence over dislocations and mechanical properties. We are sincerely grateful for your candid feedback and would greatly appreciate any further suggestions you might have regarding this work or any ongoing projects. Your continued input is valuable to us.

Best regrads,

Changfeng Wan

Reviewer 2 Report

The presented work is valuable and interesting for the reader. The authors presented the molecular dynamics simulation of the Inconel 718. The manuscript is well organized. The simulations test results extend scientific knowledge, but there are shortcomings as the following:

1.     What mean FCC? I didn’t find an expansion of the abbreviation.

2.     What mechanical properties has Inconel 718?

Author Response

Dear reviewer,

We greatly appreciate your valuable feedback on the manuscript. Our team has diligently revised the document, taking into account your insightful suggestions. Throughout the manuscript, we've incorporated specific revisions, and these are readily identifiable through highlighted text. In particular, red highlights signal reorganization that enhances both readability and logical coherence, while words added based on your comments are marked in a distinctive blue font.

Here are explanations addressing your comments:

  1. What mean FCC? I didn’t find an expansion of the abbreviation.

Response to comment: FCC means face-centered cubic which is the lattice structure of the Nickel matrix. We have included an explanation of the abbreviation "FCC" in the manuscript to ensure clarity for all readers. (See on page 2, lines 76-78, and on page 12, line 337.)

  1. What mechanical properties has Inconel 718?

Response to comment: We have included an extra introduction of the mechanical properties of Inconel 718. (e.g., the yield stress and its applicable temperature range. See on page 1, lines 21-27.)

Should you have any further inquiries or concerns, please do not hesitate to reach out. We greatly value your input and are committed to addressing reviewer feedback to enhance the quality and impact of our work.

Best regrads,

 Changfeng Wan

Reviewer 3 Report

This article discusses an investigation involving the dislocation-precipitate interaction in Inconel 718 superalloy via molecular dynamics simulations. They report that the contact position of the dislocation on the surface of the precipitate could affect the penetration depth of the dislocation. an interesting article although I have a few comments.

1. Please provide a reference for the Nose-Hoover thermostat.

2. While not the most original work, it is nonetheless interesting and should make a decent addition to the journal.

There were some grammatical errors throughout the manuscript that must be addressed.

Author Response

Dear reviewer,

We greatly appreciate your valuable feedback on the manuscript. Our team has diligently revised the document, taking into account your insightful suggestions. Throughout the manuscript, we've incorporated specific revisions, and these are readily identifiable through highlighted text. In particular, red highlights signal reorganization that enhances both readability and logical coherence, while words added based on your comments are marked in a distinctive blue font.

Here are explanations addressing your comments:

  1. Please provide a reference for the Nose-Hoover thermostat.

Response to comment: We have added a new reference to introduce the Noose-Hover thermostat. (Evans, D.J.; Holian, B.L. The Nose–Hoover thermostat. Journal of Chemical Physics 1985, 83, 4069-4074. See on page 3, line 90, and on page 13, line 394)

Should you have any further inquiries or concerns, please do not hesitate to reach out. We greatly value your input and are committed to addressing reviewer feedback to enhance the quality and impact of our work.

Best regrads,

 Changfeng Wan

Reviewer 4 Report

The article "A molecular dynamics study on the dislocation-precipitate interaction in a nickel based superalloy during the tensile deformation" is devoted to a topical issue related to the search for and improvement of new materials in aerospace, petrochemical and nuclear industries.

The authors did a good theoretical study in the article. The research results are useful and will be of interest to a wide range of readers. The quality of figures at a high level. However, the weak part of the article is the lack of elaboration of the chapter on research methodology:

1. What software was used for the simulation?

2. It is not clear from the text of the article whether only computer simulations of tensile tests were carried out or physical tests were also carried out? 

3. If physical tests were carried out, on what equipment, under what conditions, and what was the error in the simulation results compared to physical tests?

I recommend accepting the article for publication after major revisions (control missing in some experiments). 

Author Response

Dear reviewer,

We greatly appreciate your valuable feedback on the manuscript. Our team has diligently revised the document, taking into account your insightful suggestions. Throughout the manuscript, we've incorporated specific revisions, and these are readily identifiable through highlighted text. In particular, red highlights signal reorganization that enhances both readability and logical coherence, while words added based on your comments are marked in a distinctive blue font.

In light of the concerns about the quality of English writing, we've conducted a comprehensive review of the manuscript. We've focused our efforts on clarifying paragraphs that may not have been expressed with clarity. Such revisions are indicated by the presence of red font highlights in the manuscript.

Here are explanations addressing your comments:

  1. What software was used for the simulation?

Response to comment: LAMMPS is used for the simulation. We have placed additional emphasis on the software utilized for conducting the simulations within the modeling section of the manuscript. (See on page 3, line89)

  1.  It is not clear from the text of the article whether only computer simulations of tensile tests were carried out or physical tests were also carried out? If physical tests were carried out, on what equipment, under what conditions, and what was the error in the simulation results compared to physical tests?

Response to comment: While we appreciate this suggestion, the primary focus of our work lies in exploring dislocation and precipitate interactions at the nanoscale. Given the inherent challenges of conducting experiments on this scale, we have made the decision not to pursue relevant experiments in this study. We value your suggestion. Perhaps in the future, we could conduct physical experiments to observe the interactions between dislocations and precipitates and quantify their influence on the mechanical properties.

Best regards,

 Changfeng Wan

Round 2

Reviewer 1 Report

I am happy with the authors' response and recommend the article for publication.

Reviewer 4 Report

The authors answered all questions and delivered the necessary corrections to the article. Now the article can be recommended for publication (Accept in present form).